# Stromal Tissue Segmentation in Multi-Stained Serial Histopathological Sections of Pancreatic Tumors

**David Montalvo-García** [1,2]         DAVID.MONTALVO@UPM.ES
[1] *Biomedical Image Technologies Lab, Universidad Politécnica de Madrid, Spain*
[2] *CiberBBN, ISCIII, Madrid, Spain*

**Juan E. Ortuño** [2,1]           JE.ORTUNO@UPM.ES

**Ana D. Ramos-Guerra** [2,1]       ANA.RAMOS.GUERRA@UPM.ES

**Sofía Granados-Aparici** [3,4]        SOGRAAP@GMAIL.COM
[3] *INCLIVA Biomedical Health Research Institute, Valencia, Spain*
[4] *CiberONC, ISCIII, Madrid, Spain*

**Subhra S. Goswami** [1,2]         SS.GOSWAMI@UPM.ES

**Pablo Santiago Diaz** [5]        PSANTIAGODIAZ@PSMAR.CAT
[5] *Cancer Research Program, Hospital del Mar Research Institute (IMIM), Barcelona, Spain*

**Maria Evangelina Patriarca-Amiano** [5]   MEPATRIARCA@PSMAR.CAT

**Joan Lop Gros** [6]           JLOP@CLINIC.CAT
[6] *Hospital Clínic, Barcelona, Spain*

**Lidia Estudillo** [7]           LESTUDILLO@CNIO.ES
[7] *Spanish National Cancer Research Centre, Madrid, Spain*

**Mar Iglesias Coma** [5,4]        MIGLESIASC@PSMAR.CAT

**Rosa Noguera** [3,4]          ROSA.NOGUERA@UV.ES

**Nuria Malats** [7,4]           NMALATS@CNIO.ES

**María J. Ledesma-Carbayo** [1,2]      MJ.LEDESMA@UPM.ES

**Editor:** Francesco Ciompi et al.

## Abstract

In this work we propose and compare different deep learning algorithms for the segmentation of stromal regions in pancreatic histopathological image using three consecutive tissue sections, each uniquely stained with Hematoxylin and Eosin (H&E), Masson's Trichrome, and Alcian Blue. After a non-rigid registration process, variations in tissue distribution between consecutive slides still persist, which leads to distinct desired segmentations of tissues for each stain, thus underscoring the need for a specific segmentation and co-segmentation approaches to achieve higher accuracy. We compare single stain models, with respect to multi-stain techniques that either consider the multiple stains all at once in training or are based on multi-branch siamese and co-segmentation techniques. We demonstrate superior performance in identifying stromal regions with the multi-stain approaches in comparison to the segmentation techniques applied to individual stains, by effectively utilizing the complementary information each staining technique provides. This advancement is poised to enhance the further evaluation of tumor microenvironment and stromal characteristics in patients with pancreatic cancer.

**Keywords:** Segmentation, Co-segmentation, Histopathological images, Deep learning, Pancreatic cancer, H&E, Alcian Blue, Masson's Trichrome

## 1 Introduction

Stroma, the supportive tissue surrounding neoplastic cells, is not merely a passive scaffold but actively participates in tumor progression and metastasis (Grünwald et al., 2021). Accurate segmentation of stromal regions in tissue samples enables to quantitatively assess the tumor microenvironment, which is crucial for understanding tumor behavior, the interaction between cancer cells and the surrounding stroma, and the overall architecture of the tumor (Grünwald et al., 2021).

The quantification of stromal components varies markedly with the staining method used. Hematoxylin and Eosin (H&E), the most common stain used in histopathology, allows for the identification and quantification of general stromal features like cellularity and inflammatory elements, essential for assessing cell density and overall stromal morphology (Grünwald et al., 2021; Sanegre et al., 2020). Other stains offers complementary information, like Masson's Trichrome, offering detailed insights into collagen fibers, enabling the quantification of their density, distribution, and orientation, which are crucial for understanding the stiffness and architecture of the tumor microenvironment (Sanegre et al., 2020). Alcian Blue staining is key for detecting and quantifying glycosaminoglycans (GAGs) in the stroma, providing valuable information about the extracellular matrix composition and its potential influence on tumor invasion and metastasis (Sanegre et al., 2020).

This complementary analysis can be conducted by re-staining a single sample with several techniques, but this approach poses challenges, including potential alterations in tissue integrity and staining quality (Ozawa and Sakaue, 2020). As a solution, staining consecutive sections with each technique is often preferred. Yet, this method introduces its own set of complications. Notably, tissue types and structure distribution changes across sections due to micrometer-scale differences which impede perfect registrations between different stains. Therefore, applying the same segmentation across various stains is not ideal. Instead, employing distinct segmentation masks for each stain is recommended to ensure precise delineation of tissue types across all stains, enhancing the reliability and validity of the analysis.

The field of histopathology has seen significant advancements in the automatic segmentation of tissues, including stroma, using Deep Learning techniques, a development that has effectively circumvented the laborious task of manual labeling. This automation has primarily been focused on H&E stained samples, which dominate the published literature (Xu et al., 2016; Huang et al., 2017; Du et al., 2018; Al-Milaji et al., 2019). While in theory, these models could be extended to other staining methods, there exists the notable gap in the specific developme of specific methods (Huang et al., 2017; Mei et al., 2020), among other reasons due to the scarcity of labeled data and pre-trained models for stains other than H&E.

In applying advanced models to stroma segmentation in consecutively multi-stained histological samples, two paths emerge: using a distinct model for each stain or a unified model trained on all stains, yet still segmented independently. In this context Generative and self supervised methodologies are typically used to perform domain adaptation (Mei et al., 2020; Liu et al., 2024). However, there exists a third option: the co-segmentation approach, which merits consideration. Image co-segmentation refers to the process where two or more images, each containing shared objects of interest, are segmented jointly, ex-

ploiting their interdependencies and synergies. These strategies have been mainly proposed and exploited in natural imaging (Chen et al., 2019; Li et al., 2019b), and has been scarcely used in biomedical imaging (Zhong et al., 2019; Zou and Shi, 2021).

In this work we analyze the potential of different segmentation and co-segmentation approaches, which can be particularly relevant when analyzing sequential tissue sections stained with various dyes, such as H&E, Alcian Blue, and Masson's Trichrome, to comprehensively understand tissue structures and compositions in pancreatic cancer.

## 2 Methods

### 2.1 Data

We compiled data from 39 patients across two different institutions, utilizing three consecutive tissue sections per patient stained with H&E, Masson's Trichrome, and Alcian Blue or Alcian Blue PAS (Periodic Acid-Schiff), each 4 microns thick. These sections were scanned at a resolution of 0.22 microns by pixel with a Zeiss scanner. Following the non-rigid registration for precise alignment described in Section 2.2, four expert pathologists selected 3 Regions Of Interest (ROIs) of 1x1 mm from each Whole Slide Images (WSIs) for every staining technique, resulting in a total of 351 ROIs.

### 2.2 Image registration

The first step in our methodology involves the non-rigid registration of WSIs. This process is crucial for aligning the images accurately, taking into account potential deformations and inconsistencies inherent in tissue sections. The registration algorithm is based on (Gatenbee et al., 2023) and adapts to the unique characteristics of each WSI, ensuring that the ROIs in consecutive slides are precisely aligned. This alignment is essential for accurate comparison and analysis in subsequent segmentation steps. Our approach carefully considers the micro-variations in tissue structure and staining differences, ensuring that the registration is both robust and sensitive to the subtle details in the histopathological images.

### 2.3 Architectures

To address the task of stromal tissue segmentation in multi-stained serial histopathological sections, we have compared different state-of-the-art deep learning-based segmentation architectures, as well as custom architectures tailored to this problem, which incorporate the spatial information from the three serial sections stained with HE, Masson's Trichrome, and Alcian Blue or Alcian Blue PAS.

**nnUNet**. The nnU-Net framework was used as our base configuration and model as the demonstrated state-of-the-art in semantic segmentation over other architectures as visual transformers (Isensee et al., 2021, 2024). It adapts its configuration to each dataset's imaging modality and characteristics, considering image size, spacing, intensity properties, and hardware constraints, enhancing segmentation performance and generalization.

**Multi-Stain UNet**. Given the importance of considering the spatial and contextual information of the different serial sections for the segmentation of stromal tissue of each of

them, a multi-branch neural network architecture has been designed. As shown in Figure 1, this architecture receives three co-registered regions as input, one for each stain, and simultaneously segments the three regions, taking advantage of the inherent similarities and differences between them. The architecture of each branch is based on the resulting configuration of the nnUNet.

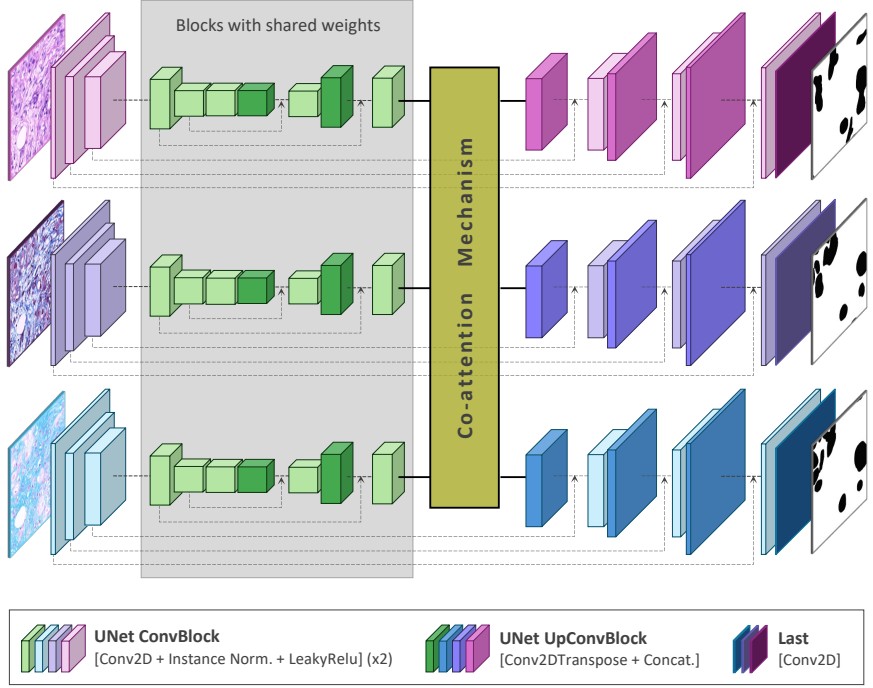

Figure 1: Multi-stain UNet, a three branch neural network architecture for the simultaneous segmentation of co-registered serial sections with different stains. The deepest blocks of the network are shared among the branches enabling the learning of features common to all stains (blocks marked as green). The number and position of shared blocks can be adjusted. The network allows the addition of a co-attention mechanism between branches.

To allow each branch the freedom to extract features specific to each stain and to construct each segmentation in the decoder, the first and last blocks of the branches are unique to the stain they represent (depicted in the Figure 1 with the representative colors of each stain). To ensure that the feature maps of the deepest blocks of the network learn features common to all three stains, certain blocks (marked in green in Figure 1) have been included following a Siamese network approach, sharing their weights between the different branches (Ji et al., 2022). The number and position of shared blocks can be varied to adjust the degree of adaptation to different stains. The architecture has been designed to include a co-attention mechanism between branches, allowing them to share contextual information extracted from the feature maps of the three stains. Among the various co-attention mechanisms available in the literature, the Co-Attention Recurrent Unit (CARU) (Li et al., 2019a) has been adapted for this problem. As shown in Figure 2, the feature maps of the three branches are fed into the recurrent neural block in the physical order in which the sections of the three stains were taken. The final output of the recurrent network is then

concatenated with each of the input feature maps. The recurrent block has a state feature map ($G_n$) that allows storing contextual information among the input feature maps. After noise removal with the reset gate ($g_z$), contextual information is extracted in the update gate ($g_d$) by combining spatial and channel attention maps using two parallel attention modules (Woo et al., 2018).

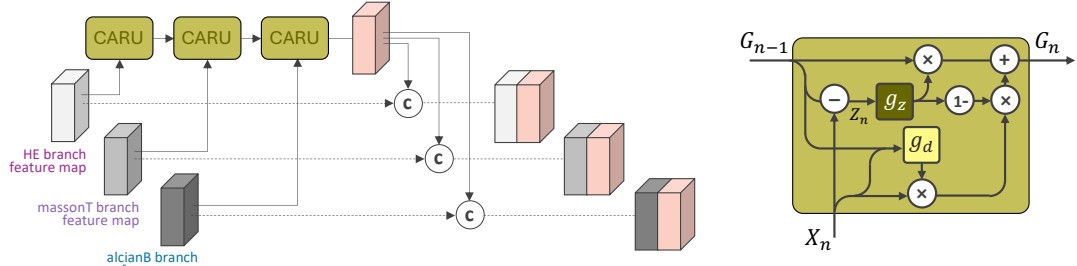

Figure 2: Detail of the co-attention mechanism that shares contextual information from the feature maps of the three stains (left) and the CARU block used (right). The state feature map ($G_n$) is denoised by a reset gate ($g_z$) and refined through an update gate ($g_d$) using spatial and channel attention modules.

## 3 Experiments and results

Of the 39 available patients, 25 were selected for training, with 5 of these used to validate the different proposed techniques, resulting in 75 ROIs for training and 15 for validation. The remaining 14 patients with a total of 42 ROIs were reserved as an independent test set.

As a first approach, an nnUNet model has been trained for each stain, encompassing the three models under the label *nnUNet_singleStain*. During inference, the corresponding model is chosen from the three available models depending on the stain to be segmented. As an alternative to using three independent models, one per stain, a single nnUNet model was trained with all stains mixed together, without discerning during training which stain was received as input. This model, labeled *nnUNet_multiStain*, showed improvements when evaluated on the validation set (see Table S1 of the supplementary material). Model hyperparameters, such as input image resolution, patch size, loss function, and network architecture, were optimized through initial experiments in the validation set. Dice and Cross Entropy was the best performing Loss function. Data augmentation techniques like rotations, mirroring Gaussian noise, Gaussian blur, brightness, contrast and gamma corrections were applied during training. The models have been trained for 400 epochs, with one epoch being defined as iteration over 250 mini-batches. Different image resolutions have been tested, and the best results have been obtained by taking patches of 1024 and halving the resolution to train the models with final patches of 512. For inference, the standard nnUNet sliding windowing procedure has been used, with window size equal to the training patch size and a 50% overlap, weighting the predictions by a Gaussian, increasing the weight of the predictions of the central patch pixels. Test time augmentation by mirroring along all axes are also applied. The models were trained on a workstation with an NVIDIA RTX A6000 (48 GB) GPU.

Regarding the multi-staining models, two alternatives have been implemented: a base model without any co-attention mechanism, with shared weights between the three stain branches in the central blocks (as displayed in Figure 1), labeled as *Siamese_UNet*, and a

second model analogous to the previous one, but adding the CARU block as a co-attention mechanism after the blocks with shared weights, labeled as *CARU_UNet*. These models have been trained with the same hyperparameters and using the same data augmentation and inference techniques detailed earlier, with three exceptions: during each epoch, all training images pass through the network; a histopathological-specific data augmentation technique has been added (Shen et al., 2022); and a further post-processing step was applied to the model output to improve the definition of the stroma tissue border, based on the morphological operations of opening and closing.

A model analogous to the *nnUNet_multiStain* model has been trained to ensure fair comparison of the multi-stain architectures with a base architecture as some small details of nnU-Net are not implemented (such as deep supervision or oversampling techniques). This model, labeled as *UNet_multiStain*, was used to pre-train the multi-stain models to account for the increased complexity and number of trainable parameters limited by the size of the training sample. This pre-trained model was used to initialize the weights of two new multi-stain models, on which fine tuning was applied to train the weights but using the co-registered input of the three stains: a first model based on Siamese nets, with all blocks shared between stains and no co-attention mechanism, labeled as *Siamese_UNet_FT*; and a second model, analogous to the previous one but with a CARU-based co-attention mechanism, keeping all the blocks shared between stains except for the block following the co-attention mechanism (since this block varies its structure as it must be able to receive as input the concatenation of the CARU output with the previous attention map of the stain). This second model has been labeled as *CARU_UNet_FT*.

To measure the algorithm's performance for each model we used the Dice score and the Intersection over Union metrics of the segmented stromal areas vs the ground truth. This metrics were assessed for all the ROIs selected by the pathologists in the validation (Table S1 of the supplementary material) and the testing datasets (Table 1). The metrics were computed with respect to the ground truth of each stain independently ("Area per Stain" in Tables S1 and 1) as well as comparing the common stromal area among the three stains in ground truth vs the corresponding resulting area for each of the models ("Common Stromal Area" in Tables S1 and 1).

| Architecture | Area per Stain | | Common Stromal Area | |
|---|---|---|---|---|
| | Dice | IoU | Dice | IoU |
| nnUNet_singleStain | $0.965 \pm 0.024$ | $0.933 \pm 0.044$ | $0.954 \pm 0.030$ | $0.913 \pm 0.054$ |
| nnUNet_multiStain | $\mathbf{0.967 \pm 0.024}$ | $\mathbf{0.938 \pm 0.044}$ | $\mathbf{0.956 \pm 0.032}$ | $\mathbf{0.917 \pm 0.057}$ |
| UNet_multiStain | $0.965 \pm 0.026$ | $0.933 \pm 0.047$ | $0.951 \pm 0.034$ | $0.909 \pm 0.060$ |
| Siamese_UNet | $0.961 \pm 0.026$ | $0.927 \pm 0.047$ | $0.949 \pm 0.032$ | $0.905 \pm 0.057$ |
| Siamese_UNet_FT | $0.965 \pm 0.025$ | $0.933 \pm 0.045$ | $0.952 \pm 0.032$ | $0.911 \pm 0.056$ |
| CARU_UNet | $0.958 \pm 0.028$ | $0.920 \pm 0.051$ | $0.944 \pm 0.040$ | $0.896 \pm 0.069$ |
| CARU_UNet_FT | $\mathbf{0.965 \pm 0.023}$ | $\mathbf{0.934 \pm 0.042}$ | $\mathbf{0.954 \pm 0.029}$ | $\mathbf{0.914 \pm 0.052}$ |

Table 1: Mean and Std. Deviation Dice and IoU metrics for the testing set ROIs, comparing models to ground truth for each stain (Area per Stain) and with respect to the Common Stromal Area among stains (Common Stromal Area).

Additionally we computed Dice score and IoU for the different stains in the testing set to assess the performance of the different models independently for each stain (Table 3 and Figure 3).

| Architecture | Dice | | | IoU | | |
|---|---|---|---|---|---|---|
| | HE | MassonT | AlcianB | HE | MassonT | AlcianB |
| nnUNet_singleStain | 0.966±0.022 | 0.972±0.019 | 0.955±0.028 | 0.936±0.040 | 0.947±0.035 | 0.916±0.051 |
| nnUNet_multiStain | **0.969±0.020** | 0.973±0.019 | **0.959±0.031** | **0.941±0.037** | **0.949±0.035** | **0.924±0.054** |
| UNet_multiStain | **0.967±0.021** | 0.972±0.018 | 0.954±0.034 | **0.937±0.038** | 0.946±0.034 | 0.914±0.058 |
| Siamese_UNet_FT | 0.967±0.022 | **0.973±0.018** | 0.955±0.030 | 0.937±0.040 | **0.948±0.033** | 0.915±0.054 |
| CARU_UNet_FT | 0.967±0.022 | **0.973±0.018** | **0.956±0.026** | 0.937±0.040 | **0.948±0.033** | **0.917±0.047** |

Table 2: Mean and standard deviation of the Dice and IoU metrics computed in the testing set ROIs for the different models with respect to the ground truth highlighting performance for each stain independently. MassonT - Masson's Trichrome. AlcianB - Alcian Blue and Alcian Blue PAS

## 4 Discussion

Results show that Muli-Stain approaches including the nnU-Net trained with all the stains perform better than the single strain approaches. Comparison of Multi-Stain methods in this data resulted in a challenging task given the good performance of all the methods and the known limitations of integrative metrics such as Dice and IoU metrics (Maier-Hein et al., 2024). Given the multiple subregions nature of stromal segmentation, distance based metrics were not considered as they could be significantly affected by segmentations that exclude small subregions, that may not be clinically relevant. The multi-stain architectures based on three branches present slightly better Dice and IoU than the base architecture *UNet_multiStain* and show less standard deviation of the metrics. Specifically the *CARU_UNet_FT* scheme is the one with higher metrics in the testing dataset showing the contribution of pre-training and co-attention mechanisms in the context of multi-stain serial segmentation. These results are also shown in the analysis per stain type where *CARU_UNet_FT* also improves the base architecture *UNet_multiStain* for Masson Trichorme and Alcian Blue based stains. In Figure 3, the top ROI shows comparable results between models, and the bottom ROI shows how CARU improves segmentation. The *nnUNet_multiStain* and the *UNet_multiStain* schemes show very good performance and could be an option easy to implement for multi stain studies with very good performance.

In this study, we have explored different multi-stain methods for segmenting consecutive regions in histopathological samples stained with different dyes. Multi-stain approaches improved single stain approaches specifically for all the stains and more notably for Alcian Blue. The variation in tissue characteristics between slides presents challenges to balance the imposed consistency vs the independent segmentations of serial slides. Some trends in the contribution of the co-segmentation techniques are observed, however metrics that allow to measure the specific gains in multi-stain studies require further research to better assess these gains and the setups in which they would be more suitable. Schemes based on self-supervised techniques and foundational models could enable the extraction of the most meaningful features from the different stains (Chen et al., 2024).

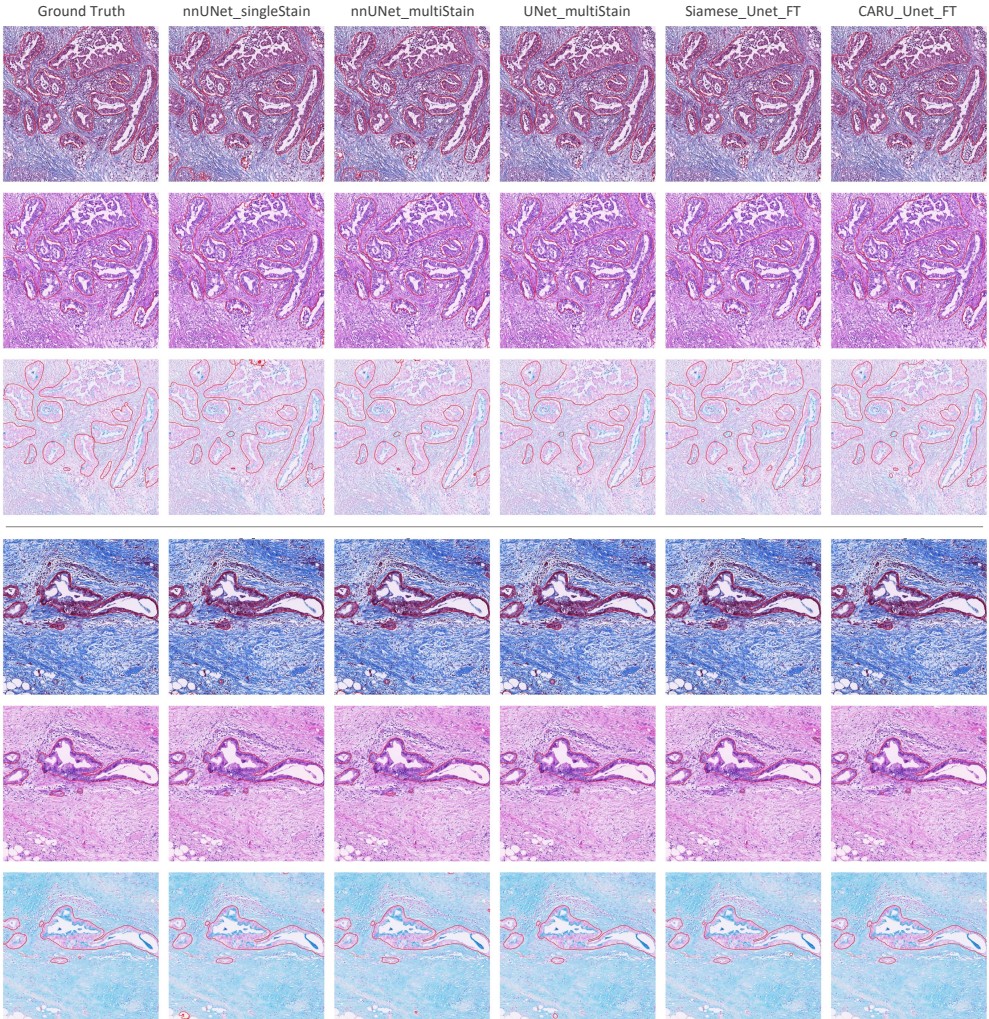

Figure 3: Examples of stromal tissue segmentation in two test ROIs with the different models compared. From left to right: Ground Truth, nnUNet_singleStain, nnUNet_multiStain, UNet_multiStain, Siamese_UNet_FT and CARU_UNet_FT.

## 5 Compliance with ethical standards

This research study was conducted retrospectively using anonymised patient data following ethical approval by the Ethics Committees of the contributing institutions and CiberONC.

## Acknowledgments and Disclosure of Funding

The authors acknowledge the support of Strategic Actions Project from CiberONC/Instituto de Salud Carlos III (ISCIII), Ministerio de Ciencia e Innovación, Agencia Estatal de Investigación, under grants PDC2022-133865-I00 and PID2022-141493OB-I00 (10.13039/5011000 11033/MCIN/AEI/ERDF, UE), the ISCIII projects INGENIO (PMP21/00107) and IM-MUNE4ALL(PMP22/00054) and the Next Generation EU funds.

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

## Supplementary Material

| Architecture | Area per Stain | | Common Stromal Area | |
|---|---|---|---|---|
| | Dice | IoU | Dice | IoU |
| nnUNet_singleStain | 0.960 ± 0.031 | 0.926 ± 0.055 | 0.946 ± 0.038 | 0.900 ± 0.067 |
| nnUNet_multiStain | **0.966 ± 0.025** | **0.936 ± 0.047** | **0.952 ± 0.034** | **0.911 ± 0.061** |
| UNet_multiStain | **0.963 ± 0.027** | **0.930 ± 0.049** | 0.948 ± 0.036 | 0.903 ± 0.064 |
| Siamese_UNet | 0.959 ± 0.030 | 0.923 ± 0.053 | 0.945 ± 0.036 | 0.897 ± 0.065 |
| Siamese_UNet_FT | **0.963 ± 0.027** | **0.930 ± 0.049** | **0.950 ± 0.035** | **0.907 ± 0.063** |
| CARU_UNet | 0.955 ± 0.032 | 0.916 ± 0.057 | 0.942 ± 0.038 | 0.892 ± 0.067 |
| CARU_UNet_FT | **0.963 ± 0.027** | **0.930 ± 0.049** | 0.949 ± 0.035 | 0.905 ± 0.062 |

Table S1: Mean and Std. Deviation Dice and IoU metrics for the validation set ROIs, comparing models to ground truth for each stain (Area per Stain) and with respect to the Common Stromal Area among stains (Common Stromal Area).

