# OpenReview forum: "Stromal Tissue Segmentation in Multi-Stained Serial Histopathological Sections of Pancreatic Tumors"
_MICCAI.org/2024/Workshop/COMPAYL — COMPAYL 2024_

### Official Review · Reviewer_252h · 2024-07-02
**good methodological comparison of several methods for stromal segmentaiton**

**Custom Rating:** 3
**Confidence:** 3

**Review:**

In this paper, the authors compared different deep learning algorithms for the segmentation of stromal regions in pancreatic histopathological image using three consecutive tissue sections, stained with H&E, MT, and AB.

Pros:
1. This is an important work on comparing different segmentation models;
2. The clinically relevant task of quantifying stromal tissue is explored;
3. Addressing a challenging task of multi-stain slides and methods to utilise the complimentary information in the stainings.

Cons:

It is hard to see any statistical difference between the compared approaches so it's hard to make a conclusion that a particular approach is better than others, especially considering that the comparison was performed only for a specific dataset and a specific cancer type, only for a single scanner. Nevertheless, it's an important step towards building knowledge through comparison studies like this.

Also, it'd be beneficial to mention the computational cost of the chosen approaches, including the number of parameters in the models, training times, as well as the inference times/costs per image/dataset.

Finally, it would be great, if possible, to release the dataset that could greatly benefit the research community for further comparison studies especially self-supervised approaches, for example.

---

### Official Review · Reviewer_6y2p · 2024-07-09
**Small performance / variance gain with shared encoder**

**Custom Rating:** 4
**Confidence:** 4

**Review:**

In general sharing an encoder to solve related tasks can lead to learning a better representation. The performance gain however is quite small. Since the dataset size is also modest, the performance delta could be due to the fact that the multistain model simply sees (significantly) more unique data. That said, the contribution is straightforward and makes sense. The argument that sharing an encoder among related tasks generally improves performance is also mostly uncontroversial.

---

### Official Review · Reviewer_QPfY · 2024-07-09
**Review of submission 25**

**Custom Rating:** 4
**Confidence:** 5

**Review:**

Summary:

The paper investigates deep learning approaches for stromal segmentation in pancreatic histopathology. It compares various models using three consecutively tissue sections with different stains (H&E, Masson's Trichrome, Alcian Blue) after non-rigid registration with VALIS. The authors propose a novel multi-branch architecture with a co-attention mechanism for multi-stain co-segmentation. Additionally, they evaluate variants like UNet vs. nnUNet, pre-training strategies, and Siamese vs. co-attention methods.


Pros:
- The paper presents an interesting approach for co-segmentation of multi-stained tissue sections.
- The paper is well written, with a clear organization.
- The proposal of a novel multi-branch architecture with a co-attention mechanism is a valuable contribution.
- The evaluation includes experiments with different model variants (UNet vs. nnUNet, pre-training, Siamese vs. co-attention).

Cons:
- While the authors claim superior performance for multi-stain approaches, Table 1 results for the test set show minimal differences between methods, particularly for nnUNet_singleStain  which differs from the best performer by 0.003. Figure 3 examples also exhibit minimal segmentation mask variations across models.
- Statistical tests are needed to assess significant differences between models.
- The dataset size appears limited. Evaluating model behavior with more training data would be beneficial.
- The models are trained using ground truth masks generated independently for the three stains. It would be interesting to conduct some experiments to check if a single ground truth mask (for example for the HE image) could be used to co-segment stroma tissue in the three sections after the registration stage.

---

### Decision · Program_Chairs · 2024-07-16

Accept